# The role of extracellular matrix phosphorylation on energy dissipation in bone

Stacyann Bailey[1]*, Grazyna E Sroga[1], Betty Hoac[2], Orestis L Katsamenis[3], Zehai Wang[4], Nikolaos Bouropoulos[5], Marc D McKee[2,6], Esben S Sørensen[7], Philipp J Thurner[8], Deepak Vashishth[1]*

[1]Department of Biomedical Engineering, Center for Biotechnology and Interdisciplinary Studies, Rensselaer Polytechnic Institute, Troy, United States; [2]Faculty of Dentistry, McGill University, Montreal, Canada; [3]Faculty of Engineering and Physical Sciences, University of Southampton, Southampton, United Kingdom; [4]Department of Mechanical, Aerospace, and Nuclear Engineering, Rensselaer Polytechnic Institute, Troy, United States; [5]Department of Material Science, University of Patras, Patras, Greece; [6]Department of Anatomy and Cell Biology, Faculty of Medicine, McGill University, Montreal, Canada; [7]Department of Molecular Biology and Genetics, Aarhus University, Aarhus, Denmark; [8]Institute of Lightweight Design and Structural Biomechanics, Vienna University of Technology, Vienna, Austria

**\*For correspondence:**
stacyann.bailey@mountsinai.org (SB);
vashid@rpi.edu (DV)

**Abstract** Protein phosphorylation, critical for cellular regulatory mechanisms, is implicated in various diseases. However, it remains unknown whether heterogeneity in phosphorylation of key structural proteins alters tissue integrity and organ function. Here, osteopontin phosphorylation level declined in hypo- and hyper- phosphatemia mouse models exhibiting skeletal deformities. Phosphorylation increased cohesion between osteopontin polymers, and adhesion of osteopontin to hydroxyapatite, enhancing energy dissipation. Fracture toughness, a measure of bone's mechanical competence, increased with ex-vivo phosphorylation of wildtype mouse bones and declined with ex-vivo dephosphorylation. In osteopontin-deficient mice, global matrix phosphorylation level was not associated with toughness. Our findings suggest that phosphorylated osteopontin promotes fracture toughness in a dose-dependent manner through increased interfacial bond formation. In the absence of osteopontin, phosphorylation increases electrostatic repulsion, and likely protein alignment and interfilament distance leading to decreased fracture resistance. These mechanisms may be of importance in other connective tissues, and the key to unraveling cell–matrix interactions in diseases.

## Introduction

In recent years, the role of extracellular matrix (ECM) proteins and their post-translational modifications (PTMs) in modulating cell activity, cell-matrix interactions, and biomineralization processes has sparked tremendous interest in different connective tissue biological systems. In particular, it has been postulated that different levels of phosphorylation of matrix proteins play a critical role in coordinating calcification processes in normally (*Murshed et al., 2004*; *Clarke, 2008*; *Murshed and McKee, 2010*; *Nudelman et al., 2010*; *Addison et al., 2010*; *Deng et al., 2013*) and pathologically (*Boskey, 2013*; *Bertazzo et al., 2013*) mineralized bone tissues either independently and/or in combination with collagen. These phosphoproteins also accumulate at the interfaces found across bone's

hierarchical levels (*McKee and Nanci, 1996*; *Thurner et al., 2009*; *Thurner, 2009*), but it remains unclear as to how their phosphorylation levels influence the mechanical properties of bone. We have recently shown the importance of phosphorylation of ECM proteins in regulating bone quality. Global phosphorylation level varied between cortical and trabecular bone (*Sroga and Vashishth, 2016*), declined with age, and was associated with an increase in age-related skeletal fragility (*Sroga and Vashishth, 2018*).

Given the importance and incomplete understanding of how total phosphorylation levels, as well as the heterogeneity of phosphorylation observed for different bone matrix proteins, contribute to skeletal fragility, animal models provide a valuable resource to investigate this further. In particular, certain animal models recapitulate key metabolic and skeletal characteristics seen in humans displaying, for example, the phenotypes of major phosphate-handling diseases such as **hypo**phosphatemia (*Barros et al., 2013*; *Liu et al., 2006*; *Boukpessi et al., 2017*), *hyper*phosphatemia (*Yuan et al., 2014*), and hypophosphat*asia* (*Harmey et al., 2006*; *Narisawa et al., 2013*; *Yadav et al., 2014*). Hyp mice – the murine analog of X-linked hypophosphatemia (XLH) – display low-serum phosphate and accumulation of osteopontin (OPN) (*Barros et al., 2013*), a well-known noncollagenous protein serving as a powerful inhibitor of mineralization, and a key determinant of bone's resistance to fracture. In contrast to Hyp mice, fibroblast growth factor 23-deficient mice (*Fgf23*[-/-] mice) are hyperphosphatemic, but like the Hyp mice also show accumulation of OPN (*Yuan et al., 2014*). Both of these phosphate-handling disease models exhibit a soft-bone (osteomalacia) phenotype and display decreased cortical area, thickness, and strength (*Liu et al., 2016*; *Murali et al., 2016*). The hypophosphatasia mouse model (*Alpl*[−/−] mice) displays mineralization deficiencies characterized by rickets/osteomalacia as well as elevated levels of inorganic pyrophosphate (PPi). The *Alpl*[−/−] mice also show increased levels of phosphorylated OPN compared to wild type (WT) mice (*Narisawa et al., 2013*). Interestingly, Opn KO mice also show elevated levels of PPi despite having more mineralized osteoid than wildtype (WT) controls (*Harmey et al., 2006*). As such, it appears that OPN levels, and possibly its phosphorylation status, contribute to impaired matrix mineralization and may play a role in skeletal integrity in these models.

The degree of OPN phosphorylation has significant effects on its structure and physiological function (*Kazanecki et al., 2007*). For example, osteoclast adhesion is increased with phosphorylation (*Katayama et al., 1998*) and correlates with the extent of bone resorption (*Razzouk et al., 2002*). Hydroxyapatite (HA) crystal formation and growth are inhibited by OPN in a dose-dependent manner (*Hunter et al., 1994*), and dephosphorylation of OPN abolishes the inhibitory effect of OPN on HA formation by at least 40-fold (*Razzouk et al., 2002*). In addition to bone resorption and mineralization, OPN has been shown to play a mechanical role in bone, influencing its resistance to fracture (*Duvall et al., 2007*; *Thurner et al., 2010*; *Poundarik et al., 2012*). The negatively charged phosphate groups of serine and threonine residues on OPN bind to multivalent positive ions on hydroxyapatite, and this interaction is part of a bonding/cohesion process that limits separation of mineralized collagen fibrils during mechanical loading (*Zappone et al., 2008*). Also important in this bone-toughening process are large, covalently crosslinked (by transglutaminase) networks formed between neighboring OPN molecules and between OPN and other bone matrix proteins (*Kaartinen et al., 1999*; *Kaartinen et al., 1997*; *Kaartinen et al., 2002*). Networks of crosslinked OPN polymers are abundantly present in bone, and may reside in the interfibrillar collagenous matrix, at cell-matrix interfaces, and in interfacial cement lines (*Kaartinen et al., 2002*; *Goldsmith et al., 2002*) where they may be critical for maintaining the overall strength of bone tissue (*Kaartinen et al., 1999*; *Hoac et al., 2017*; *Cavelier et al., 2018*). Analysis of different tissues revealed that phosphorylation of OPN is highly variable, and typically only some of the potential phosphorylation sites are occupied in vivo (*Neame and Butler, 1996*). In fact, it is currently unknown how many of all available amino acid residues in mouse OPN are phosphorylated in vivo because the balance between the activities of protein kinases and phosphatases reflects the phosphorylation state of the protein. Importantly, the difference in phosphorylation status results in altered biological and mechanical responses. Considering the functional relevance of OPN phosphorylation, this PTM may be an important determinant of bone matrix quality and fragility.

In this study, we investigate the role of OPN phosphorylation on bone fracture. To execute this, we first demonstrate that in mouse models of impaired phosphate regulation and increased skeletal fragility, the level of OPN phosphorylation declines. Next, we captured the effects of phosphorylated OPN on bone fracture toughness (resistance to crack propagation and fracture) by developing

methods to enzymatically phosphorylate and dephosphorylate WT and OPN-deficient mouse bones ex vivo, then measure the resultant change in their mechanical competence. In an effort to gain a better understanding of the various factors that contribute to the mechanical function of phosphorylated OPN in bone matrix, we then conducted atomic force microscopy-force spectroscopy (AFM-FS) experiments demonstrating the effect of pH, ion charges, and phosphorylation levels on the energy dissipation properties of the OPN network using simplified synthetic and physiologically relevant surfaces. Based on these results, we propose that for appropriate mechanical function of bone, the phosphorylation status of OPN promotes fracture toughness up to a beneficial point. Phosphorylation or dephosphorylation alters the interaction between charged groups on OPN, and between OPN and bone mineral leading to increased or decreased energy dissipation.

## Results

### Evidence of decreased osteopontin phosphorylation in mouse models of impaired phosphate metabolism and decreased mechanical properties

We first investigated whether the phosphorylation state of OPN varied using in vivo mouse models having phosphate disorders and known skeletal pathology linked to soft osteomalacic bones. Mineral-bound proteins were extracted from long bones of WT, Hyp, and $Fgf23^{-/-}$ mice. Total protein was quantified using a colorimetric detection system. From each sample, 2 µg of protein extract was loaded onto a 4–20% gradient SDS-PAGE gel. Since the vast majority of OPN phosphorylation occurs at serine residues, we performed immunoblotting for phosphoserine in the mineral-binding protein extracts. In the bone matrix of both Hyp and $Fgf23^{-/-}$ mice, we found that mineral-bound OPN increased (*Figure 1a*) but global phosphorylation decreased (*Figure 1b*) as compared to WT controls. In addition, the post-immunoprecipation results show that despite the accumulation of OPN in Hyp and $Fgf23^{-/-}$ mice (*Figure 1c*), the proportion of phosphorylated OPN was reduced compared to the bone of WT mice (*Figure 1d*). Given that these models have opposite levels of serum

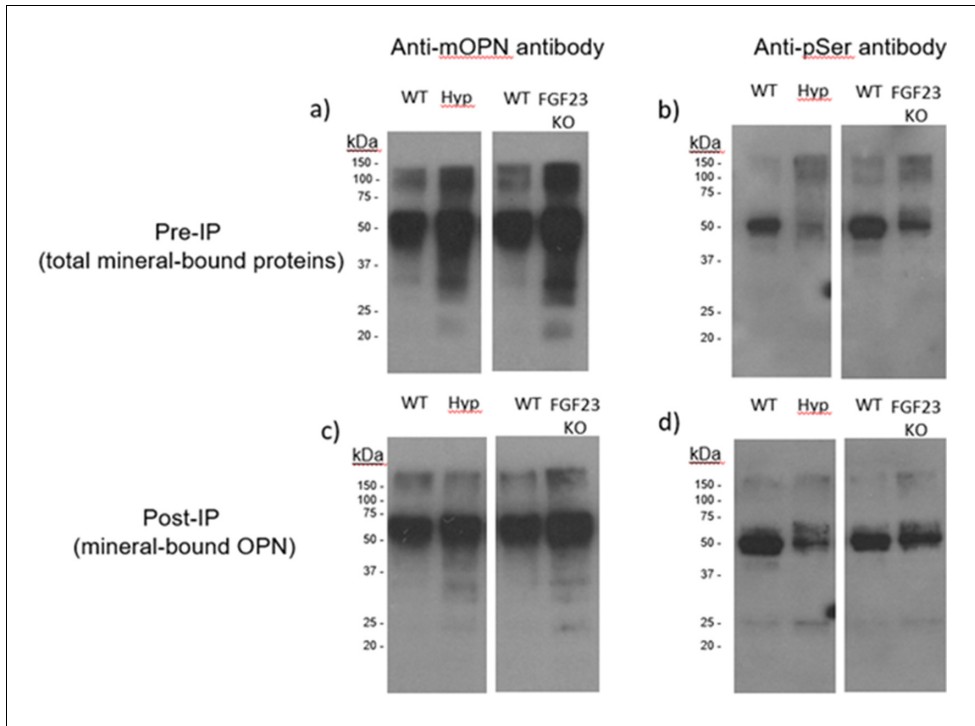

**Figure 1.** Pre-immunoprecipation (Pre-IP) of mineral-bound OPN. (**a**) and global phosphorylation (**b**) in protein extracts of long bones from WT, *Hyp* and *Fgf23*$^{-/-}$ mice. Post-immunoprecipation (Post-IP) indicates that despite similar levels of OPN (**c**), phosphorylation of OPN is reduced in these disease models (**d**).

phosphate deviation from normal (hypophosphatemic vs. hyperphosphatemic), and display a reduction in bone strength (*Liu et al., 2016*; *Murali et al., 2016*; *Sitara et al., 2004*; *Camacho et al., 1995*) which may be dependent on defective mineralization but driven by the mineral-inhibiting protein OPN, our results suggest that osteopontin phosphorylation may be an important contributor to the fracture resistance of bone.

## Phosphorylation status of osteopontin influences bone fracture toughness

To capture the effects of OPN phosphorylation on bone fracture toughness, we performed separate ex-vivo phosphorylation and dephosphorylation of whole femurs from WT and Opn KO mice and subsequent mechanical testing. The global phosphorylation level in bone matrix increased in both genotypes with ex-vivo casein kinase-II (CKII) phosphorylation (WT-*phosphorylated vs.* WT-*nonphosphorylated control, p=0.008; Opn KO-phosphorylated vs. Opn KO-nonphosphorylated control, p=0.007*) (*Figure 2a*). We observed a significant reduction in phosphoproteins with ex-vivo dephosphorylation by alkaline phosphatase (WT-*dephosphorylated vs.* WT-*nondephosphorylated control, p=0.033; Opn KO-dephosphorylated vs. Opn KO-nondephosphorylated control, p=0.006*) (*Figure 3a*). Although the change in ex-vivo phosphorylation between WT and Opn KO (delta-WT vs. delta-Opn KO, *Figure 2b*) was not statistically significant, we observed a significant difference in dephosphorylation between delta-WT and delta-Opn KO (*Figure 3b*), indicating that OPN-deficient bone can be modified to a greater extent than WT bone, likely attributable to increased permeability of enzymes into bones lacking OPN.

We observed higher fracture toughness with phosphorylation of WT bones (*WT-phosphorylated vs. WT-nonphosphorylated control, p=0.009*). In contrast, toughness declined in Opn KO mice following ex-vivo phosphorylation (Opn KO-*phosphorylated vs.* Opn KO-*nonphosphorylated control, p=0.025)* indicating that phosphorylation of other bone matrix proteins in the absence of OPN did not improve the fracture resistance of bone (*Figure 4*). Ex-vivo dephosphorylation caused a decrease in fracture toughness for the WT group (WT-*dephosphorylated vs.* WT-*nondephosphorylated control, p=0.012*) (*Figure 5*) while an increase in fracture toughness was observed for Opn KO mice (Opn KO-*dephosphorylated vs.* Opn KO-*nondephosphorylated control, p=0.037*).

## Energy dissipation of the osteopontin network is altered by levels of phosphorylation

We conducted atomic force microscopy-force spectroscopy (AFM-FS) studies using an in-vitro experimental system to demonstrate that the phosphorylation status of OPN can affect bone toughness by altering energy dissipation. At pH 8.5, both OPN and hydroxyapatite (HA) surfaces are negatively charged. Under high $Ca^{2+}$ concentration, the detachment energy increased more significantly as compared to $H_2O$ and $Na^+$ environments. At pH 6.0 however, the protein and HA bear opposite charges. The slightly acidic pH lead to a moderate dissolution of HA over time, and therefore, it is expected that $Ca^{2+}$ ions are present in solution from the beginning of the experiment. Further addition of $Ca^{2+}$ ions decreased energy dissipation attributable to the reduction of sacrificial bond

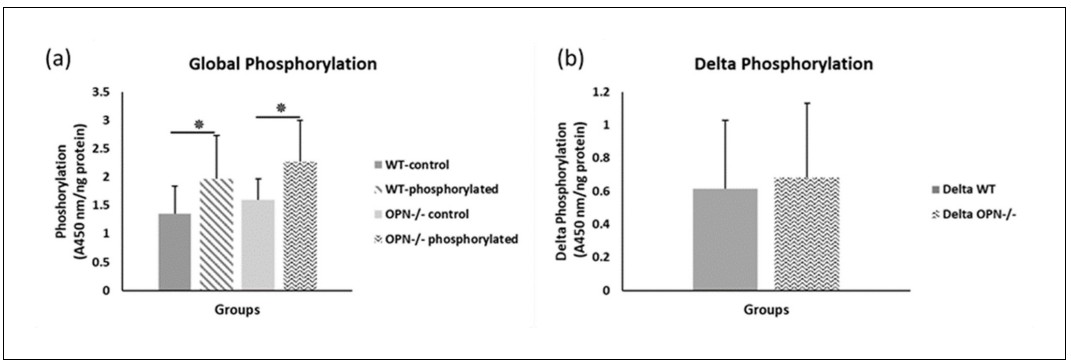

**Figure 2.** Mean global protein phosphorylation. (a) and change in phosphorylation (b) for WT and Opn KO groups. * indicates significance at p<0.05 and error bars represent standard deviation.

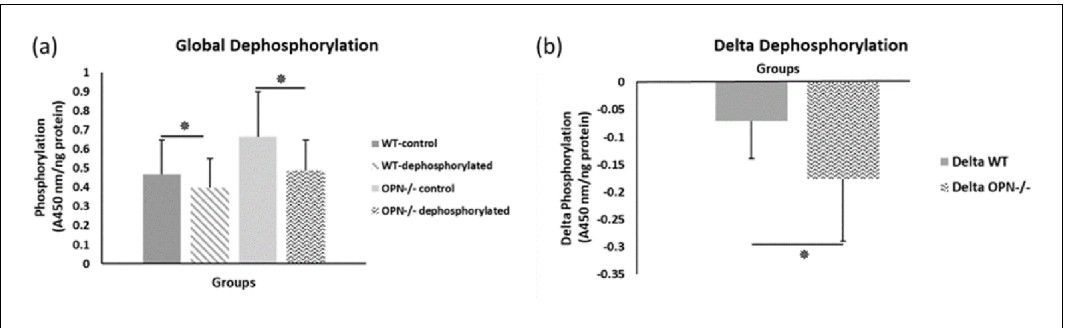

**Figure 3.** Mean global protein phosphorylation. (**a**) and change in phosphorylation (**b**) after removal of phosphate groups (dephosphorylation) for WT and Opn KO groups. * indicates significance at p<0.05 and error bars represent standard deviation.

formation (*Gao et al., 2003*) (increase of effectively positively charged sites in OPN) as well as increased repulsion between OPN and HA. These results were also confirmed by the decline in energy dissipation that was observed for dephosphorylated OPN in $Ca^{2+}$ solution at pH 7.4 as compared to native OPN, as well as dephosphorylated OPN in $Na^+$ solution at pH 7.4, all on mica substrates. Thus, the balance between $Ca^{2+}$ ions in solution and the availability of negatively charged groups are both important for energy dissipation within the OPN network as well as at the OPN-HA interface. The results from AFM force spectroscopy are summarized in *Figure 6*.

## Discussion

Extracellular bone matrix phosphoprotein osteopontin (OPN) has been recently implicated in disease models of *hypo*phosphatemia (*Barros et al., 2013*; *Liu et al., 2006*; *Boukpessi et al., 2017*), *hyper*-phosphatemia (*Yuan et al., 2014*), and/or hypophosphat*asia* (*Harmey et al., 2006*; *Narisawa et al., 2013*; *Yadav et al., 2014*). Consistent with these studies, we observed full-length OPN in bone extracts of Hyp and *Fgf23$^{-/-}$* mice. We further provide evidence that the phosphorylation level of OPN declined in these mouse models as detected by immunoblotting for OPN's phosphoserine residues. Given that these models display an aging-like skeletal phenotype (*Sroga and Vashishth, 2018*) with impaired mineralization and osteomalacia, we considered whether the phosphorylation status of bone matrix proteins including OPN is an important determinant of their skeletal fragility. Our experimental model involved phosphorylation and dephosphorylation of both normal WT bones with OPN, and bones without OPN (Opn KO); thus, by comparing the change in fracture toughness caused by phosphorylation or dephosphorylation of the organic matrix between these two samples (WT-treated minus WT-control, delta-WT; and Opn KO -treated minus Opn KO -control, delta- Opn

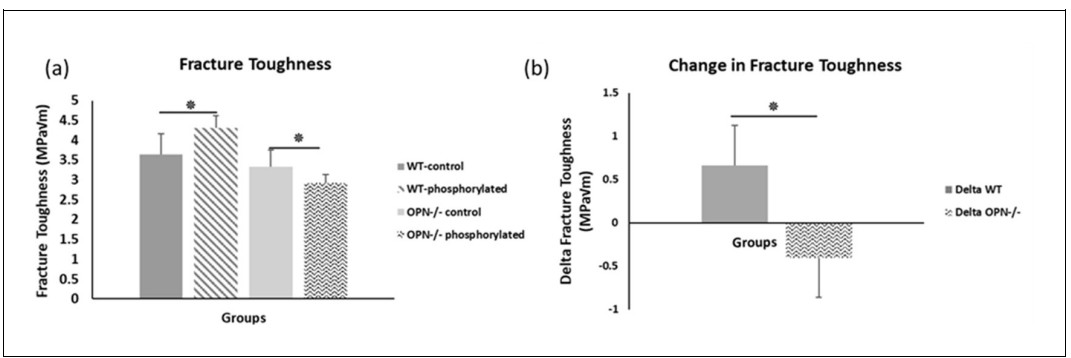

**Figure 4.** Mean fracture toughness (**a**) and change in fracture toughness (**b**) due to ex-vivo phosphorylation for WT and Opn KO groups. * Indicates significance at p<0.05 and error bars represent standard deviation.
The online version of this article includes the following source data for figure 4:

**Source data 1.** Fracture toughness of phosphoryled WT and Opn KO mice.

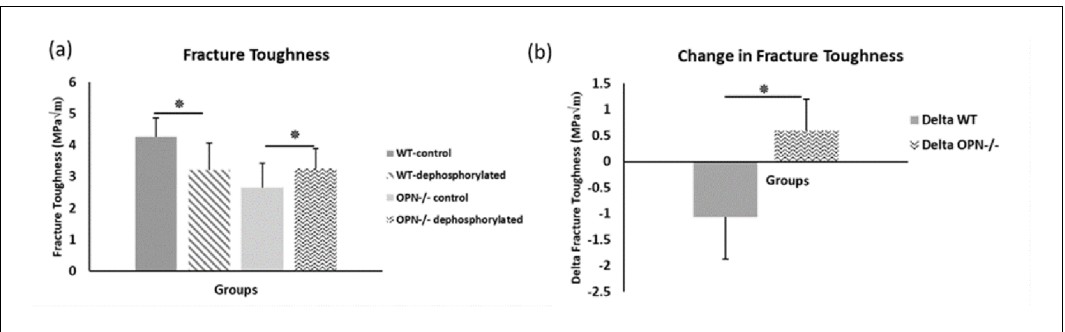

**Figure 5.** Mean fracture toughness (**a**) and change in fracture toughness (**b**) attributable to ex-vivo dephosphorylation for WT and Opn KO groups. * Indicates significance at p<0.05 and error bars represent standard deviation.

The online version of this article includes the following source data for figure 5:

**Source data 1.** Fracture toughness of dephosphoryled WT and Opn KO mice.

KO), the contribution of OPN phosphorylation alone can be isolated. Our results suggest that OPN and its phosphorylation level may be one of the dominant phosphoproteins in the determination of global bone matrix phosphorylation level and bone fracture toughness.

Fracture resistance of bone emerges from various mechanisms that exists at multiple length scales across bone hierarchy, and involves in part growth and packing of mineral foci into larger crossfibrillar aggregates such that the ECM becomes highly mineralized. In the context of the present work, at the nanoscale, major contributions to the intrinsic toughness of bone originate from the OPN-crosslinked protein networks (*Cavelier et al., 2018*; *Fantner et al., 2007*) and the formation of dilatational bands involving osteocalcin (OC)-osteopontin complexes (*Poundarik et al., 2012*). The OC-OPN complex has been recently shown to provide high shear toughness and ductility to the interfibrillar interface (*Wang et al., 2020*). Both the OPN-crosslinked protein networks and the OC-OPN complex presumably work together to control deformation and separation of mineralized collagen fibrils (*Gao et al., 2003*; *Zimmermann et al., 2012*). Here, we propose two co-existing mechanisms to elucidate how the addition or removal of phosphate groups on proteins, and particularly OPN, could affect bone mechanical function.

First, cation-mediated crosslinks are formed between two binding regions on one OPN polymer, multiple OPN polymers, and OPN and charged surface ions on HA (*e.g.*, $Ca^{2+}$, $Na^+$) (*Figure 6—figure supplement 3*; *Fantner et al., 2005*). These salt-bridges are weak, but reformable sacrificial bonds that prevent portions of OPN polymers from rupturing (cohesion of the OPN meshwork) and debonding of OPN from HA during repetitive mechanical loading (*Zappone et al., 2008*; *Fantner et al., 2007*; *Lai et al., 2014*). The high affinity of OPN to $Ca^{2+}$ ions was reinforced in our AFM-FS studies. We used bovine milk OPN as the model protein because of its natural and extensive phosphorylated status (*Sørensen et al., 1995*). In the presence of $Ca^{2+}$ ions and when both phosphorylated milk OPN and HA are negatively charged, detachment energy increased significantly. The increase in detachment energy was also observed between OPN and mica substrate (*Figure 6*). $Ca^{2+}$-mediated crosslinks were also formed between OPN polymers, which increased cohesion of the OPN meshwork, indicated by higher detachment energy (*Figure 6—figure supplement 2*), which is generally associated with loading of multiple molecules in parallel (*Fantner et al., 2006*). Thus, via the effects mentioned above the meshwork is able to stretch more and increase the energy required for complete detachment. Dephosphosphorylation of milk OPN or reversing the charge on HA both resulted in decreased energy dissipation (*Figure 6*). Similarly, phosphorylation of WT bone specimens ex-vivo under our experimental conditions caused an approximate 18% increase in fracture toughness (*Figure 4a*) whereas, dephosphorylation decreased toughness by 25% (*Figure 5a*). These results suggest that phosphorylation is enabling various matrix/mineral interactions, and hence, dissipating energy.

Second, phosphorylation can alter protein network conformation, the mechanical behavior of the organic matrix, and consequently the macroscopic fracture toughness of bone (*Thurner et al., 2009*; *Fantner et al., 2007*; *Fisher et al., 2001*). A recent experimental study (*Malka-Gibor et al., 2017*)

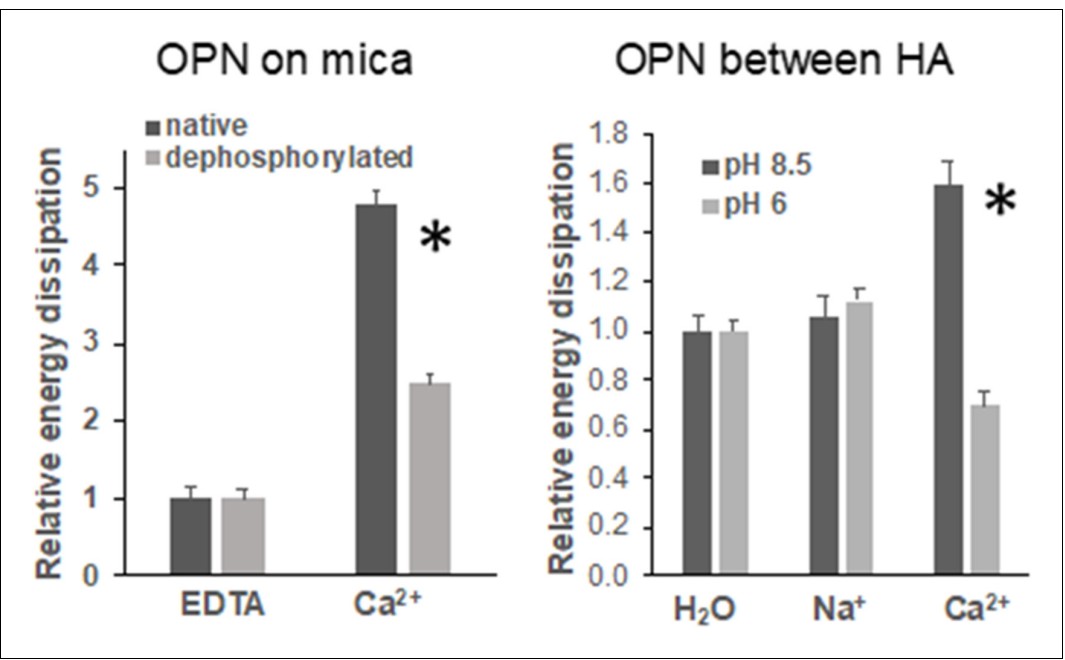

**Figure 6.** Energy dissipation of OPN networks during AFM-FS experiments. Energies are normalized to dissipation levels in EDTA for OPN deposited on mica and pulled with a pristine AFM tip (pH 7.4) and to dissipation levels in $H_2O$ for OPN deposited on HA and pulled with a HA-functionalized tip. All values are significantly different except OPN between HA, pH 8.5 $H_2O$ vs. $Na^+$. It should be noted that the relative differences are similar to what is seen for quantitative values, except for EDTA and $H_2O$ levels due to normalization. These values are provided in *Supplementary files 1* and *2*. * indicates significance at $p < 0.05$ and error bars represent standard error (SE) of the mean.

The online version of this article includes the following source data and figure supplement(s) for figure 6:

**Source data 1.** Energy dissipation of native (phosphorylated) and dephosphorylated OPN film on mica in EDTA and calcium solution.

**Source data 2.** Energy dissipation of native (phosphorylated) OPN film on HA under various pH and ionic conditions.

**Figure supplement 1.** Backscattered electron image of an AFM probe and HA surface.

**Figure supplement 2.** Representative force spectroscopy curve of hydrated OPN.

**Figure supplement 3.** Proposed model for the OPN-HA interaction in different ionic- and pH environments.

demonstrated that intrinsically disordered proteins (IDPs) and their phosphorylation status can alter neurofilament protein alignment and distance between filaments, resulting in changed energy dissipation of the network (*Figure 7*). Neurofilaments are a valuable model system for examining phosphorylation-driven interactions of IDPs owing to their high modularity in protein content and phosphorylation levels. Both collagen and neurofilaments are bundled network systems that interact with IDPs (*Laser-Azogui et al., 2015*; *Orgel et al., 2006*). For example, non-collagenous proteins in bone matrix such as small integrin-binding ligand, N-linked glycoproteins (SIBLINGs) are IDPs, interact with collagen, and gain more folded features when post-translationally modified (phosphorylation, glycation, acetylation, sulfation, cleavage) (*Boskey and Villarreal-Ramirez, 2016*). In this regard, the SIBLING proteins (e.g. osteocalcin, osteonectin, fibrillins, etc.) interacting with collagen filaments/fibrils may be considered analogous to neurofilament proteins (*Laser-Azogui et al., 2015*; *Yuan et al., 2017*; *Khalil et al., 2018*). Our AFM-FS studies showed that in the presence of excess $Ca^{2+}$ ions strong cohesion and excessive crosslinking of the OPN meshwork reduces the stretching ability of the meshwork, leading to shorter pulls, increased repulsion of all positive sites, which likely increased distance in the meshwork, and diminished detachment energy. We postulate that the increase in global phosphorylation of other bone matrix proteins in the absence of OPN ( e.g. other SIBLING matrix proteins) may also potentially result in increased protein alignment and larger interfilament distance between mineralized collagen fibrils, to a detrimental degree that decreases matrix

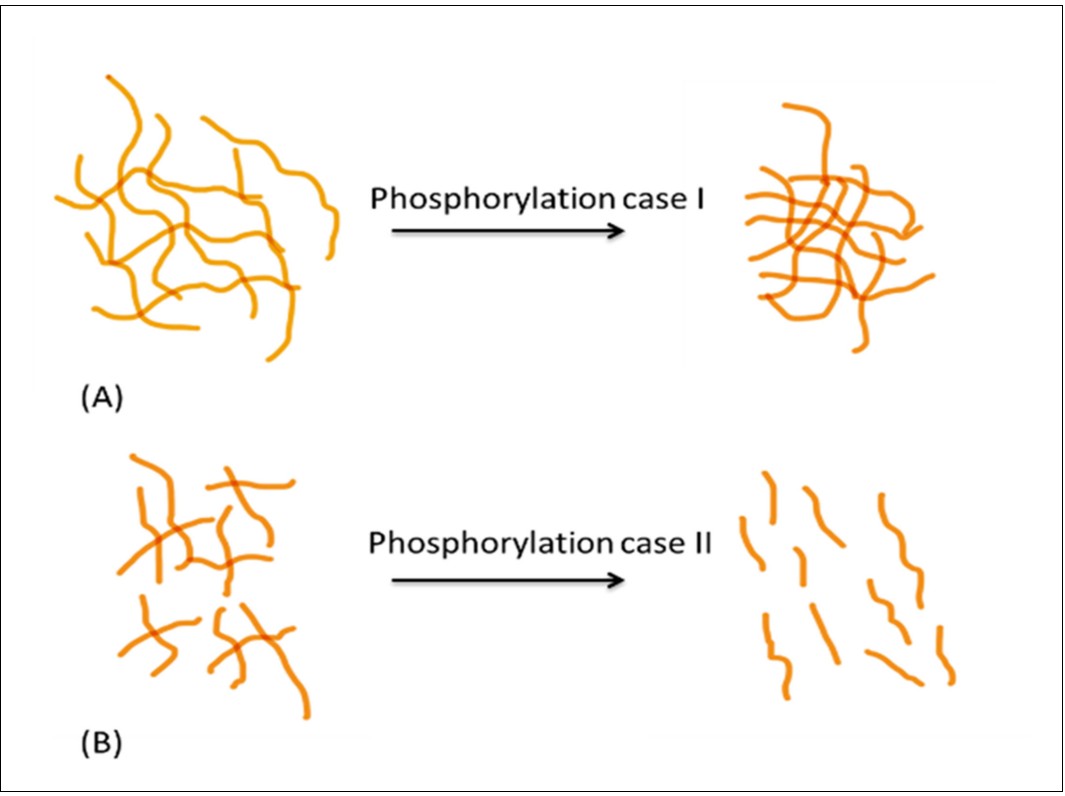

**Figure 7.** Schematic diagram showing differential effects of phosphorylation on conformation of protein systems. In protein system (**A**), phosphorylation tends to increase inter- and intrafilament interactions, hence the interfilament distance is reduced. In protein system (**B**), phosphorylation tends to create interfilament repellant, hence increasing the protein system alignment and inter- filament distance.

interaction, energy dissipation, and consequently fracture resistance in osteopontin-deficient mice (*Figure 4*).

We observed a non-linear dose response relationship between the level of global matrix phosphorylation and bone fracture toughness in WT mice (*Figure 8a*). Phosphorylation explained ~36% of the variance in fracture toughness and this relationship was not observed in the absence of OPN (*Figure 8b*). Taken together, this data supports the previously mentioned mechanism involving increased interaction energy and sacrificial bond formation between OPN and HA as well as between OPN polymers. The AFM-FS studies show that adhesion is not only dependent on the charge of OPN and HA under a certain environment but also the availability of free $Ca^{2+}$ ions. The remaining variance in fracture toughness may be associated with the formation of OC-OPN complexes or enzymatic OPN-crosslinked protein networks. It has been previously shown that crosslinking of OPN by transglutaminase-2 enzyme (TG2) increases interfacial adhesion and toughness (*Cavelier et al., 2018*). However, OC inhibits TG2 crosslinking activity most likely by competing for the binding site on OPN (*Kaartinen et al., 1997*). As such, there is insufficient evidence at present that TG2 crosslinking of OPN and phosphorylation of OPN are independent. Although ex-vivo phosphorylation of Opn KO mice bone decreased fracture toughness (*Figure 4*), and dephosphorylation increased toughness compared to the respective untreated Opn KO mice bone (*Figure 5a*), unlike WT, we did not observe an association between the level of global matrix phosphorylation and fracture toughness in these mice (*Figure 8b*). As noted above, excessive crosslinking can be detrimental to protein networks by increasing repulsion, interfilament distance, and stretching ability. This data suggests that although global matrix level of phosphorylation affects fracture toughness, the contribution of phosphorylated OPN may be critical in the determination of bone toughness.

Our current study is not devoid of limitations and we acknowledge other phosphorylation interactions that may potentially influence the outcomes. The gross skeletal phenotype of Opn KO mice is normal compared to WT mice (*Rittling et al., 1998*; *Yoshitake et al., 1999*). However, increased

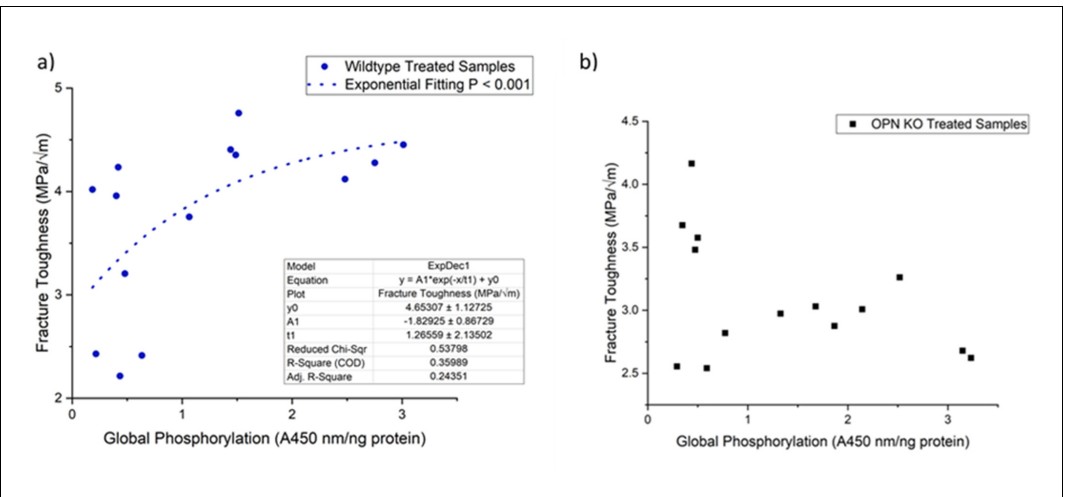

**Figure 8.** Schematic of the relationship between global protein phosphorylation and fracture toughness of wild-type (**a**) and Opn KO (**b**) mice. By continuing the increase in phosphorylation of WT bone, fracture toughness improves exponentially. There is no significant relationship between global phosphorylation and fracture toughness in Opn KO mice following ex-vivo phosphorylation and dephosphorylation.

mineralization was found in some areas of cortical bone (*Boskey et al., 2002*), and the bones are mechanically weaker. The collagen structure in Opn KO mice was also shown to be highly disorganized which further causes disorganization of mineral (*Depalle et al., 2020*). OPN in bone resides at its surfaces (including lining the lacuno-canalicular system) in the thin structure known as the *lamina limitans* (*McKee and Nanci, 1996*), and throughout bulk bone. Thus, its alterations in vivo may affect many processes including mineral-binding (*Addison et al., 2010*), cell attachment as part of the bone remodeling cycle, cell signaling that may affect mechanosensation, and the structural integrity of bone. Our ex-vivo experiments were conducted under physiological conditions to alter the organic matrix with buffer solutions containing magnesium chloride, calcium, and EDTA to prevent any alterations in mineral. The AFM measurements are not fully quantitative, but the potential lies in examining relative differences, as was done in this study. Also, by using the same cantilever, the measurements are very accurate and reproducible. Bovine milk OPN contains approximately 28 phosphorylation sites and all but a few residues in this motif are phosphorylated. The higher phosphorylation levels essentially allow for demonstration of the principal effects seen in whole-bone fracture toughness tests following increased phosphorylation. Attempts to over-phosphorylate bovine milk OPN (our source OPN) would likely be unsuccessful as the nonphosphorylated serine residues in bovine OPN are not located in recognition sequences of any specific kinase. The lack of experiments on OPN with a varying range of phosphorylation levels may be seen as a limitation, but nevertheless we provide data points for the most extreme cases, and, different from the physiological system, we control the concentration of $Ca^{2+}$ ions.

Despite the fact that the maximum-load method for measuring fracture toughness demonstrates the least variability compared to other methods (*Ritchie et al., 2008*), there is inherent variability in fracture toughness tests. For example, we have shown with a larger sample size that Opn KO mice have lower fracture toughness compared to WT mice (*Thurner et al., 2010*; *Poundarik et al., 2012*). The data in *Figures 4* and *5* are from a different set of control bones and fracture toughness values vary across bones from the same batch of mice attributable to inherent differences between animals, and because of variations in any mechanical testing method (including fracture toughness testing). Accordingly, we have minimized variations between animals by conducting pairwise comparison i.e. WT-dephosphorylated vs WT-controls. Such comparison, as noted above, also allows us to determine the contribution of OPN phosphorylation and dephosphorylation while accounting for compounding effects of other changes in the bone matrix.

In conclusion, this study shows for the first time that osteopontin and its phosphorylation level promotes fracture toughness of bone. The heterogeneity in osteopontin phosphorylation, alters interfacial adhesion and cohesion of the OPN meshwork leading to increased or decreased energy

dissipation. In the absence of osteopontin, phosphorylation and de-phosphorylation of other bone matrix proteins impact bone toughness in a binary stepwise manner. We expect that our study holds the potential to begin understanding the need for regulation of global matrix phosphorylation and heterogeneity in phosphorylation for different proteins with respect to maintaining skeletal health and whose alterations influence bone fragility in diseases.

# Materials and methods

## Key resources table

| Reagent type (species) or resource | Designation | Source or reference | Identifiers | Additional information |
|---|---|---|---|---|
| Genetic reagent (*M. musculus*) | C57BL/6NCrl | Charles River | RRID:IMSR_CRL:27 | |
| Genetic reagent (*M. musculus*) | B6.Cg-*Phex^Hyp*/J | Jackson Laboratory | Cat#: 000528 RRID:IMSR_JAX:000528 | Animals maintained in Dr M Mckee lab. |
| Genetic reagent (*M. musculus*) | *Fgf23^-/-* | PMID:15579309 | | Animals were a gift from Dr. B. Lanske |
| Genetic reagent (*M. musculus*) | B6.129S6 (Cg)-*Spp1^tm1Blh*/J | PMID:9661074 | | Animals were a gift from Dr S. Rittling. |
| Genetic Reagent (*B. taurus*) | Milk protein (Mammary gland) | PMID:8320368 | | Provided by Dr ES Sorensen |
| Chemical compound, drug | Synthetic hydroxyapatite | *Andriotis et al., 2010*. Crystal Research and Technology | | Produced by Dr N. Bouropoulos |
| Commercial assay or kit | pIMAGO-biotin HRP Detection | Tymora Analytical | Cat# 900–100 | |
| Antibody | anti-OPN (goat polyclonal) | R and D Systems | Cat# AF808, RRID:AB_2194992 | (1:100,000 µL) |
| Antibody | anti-phosphoserine (rabbit polyclonal) | Thermo Fisher Scientific | Cat# 61–8100, RRID:AB_2533940 | (1:2500 µL) |

## Immunoprecipitation and immunoblotting for OPN in mouse models of phosphate disorder

Long bones from 6-week-old Hyp and *Fgf23^-/-* mice (and WT age-, strain-, and sex-matched controls, n = 3) were collected and bone proteins extracted as described previously (*Goldberg and Sodek, 1994*). In brief, cleaned frozen bone samples were pulverized, cooled in liquid nitrogen, and bone protein extracted from this powder twice at 4°C for 24 hr with 4 M guanidium-HCl in 50 mM Tris-HCl, pH 7.4 containing protease and phosphatase inhibitors (0.1 mM phenylmethylsulfonyl fluoride (PMSF), 100 µg/mL of benzamidine, 5 µg/mL leupeptin, 1 mM sodium pyrophosphate, 1 mM β-glycerophosphate, 1 mM sodium orthovanadate, and 5 mM sodium fluoride). Mineral-bound proteins were then extracted twice at 4°C for 24 hr with 0.5 M EDTA, 50 mM Tris-HCl, pH 7.4 containing protease and phosphatase inhibitors. The mineral-binding protein fraction was then concentrated and washed in 5 mM sodium bicarbonate, then quantified using the bicinchoninic acid protein assay (Pierce, Rockford, IL, USA).

For each sample, 10 µg of total mineral-bound bone protein extract was mixed with 300 µL of 100 mM sodium acetate, pH 5.5 containing 1 mM PMSF and 0.1 mM leupeptin and incubated on ice for 3 min, and then gently mixed with rotation at 4°C for 10 min. Next, 10 µL of 0.2 mg/mL goat anti-mouse osteopontin antibody (R and D Systems, Cat# AF808-CF, Minneapolis, MN, USA) was added and samples were gently rotated at 4°C for 1 hr, followed by the addition of 50 µL of Protein A/G PLUS-Agarose beads (Santa Cruz, SC-2003, Dallas, TX, USA) and gentle rotation at 4°C for 1 hr. Samples were spun at 2000 × g for 1 min, and supernatants were removed. Beads were then washed in cold 100 mM sodium acetate buffer, pH 5.5 three times, and immunoprecipitated proteins were eluted in 2 × Laemmli protein loading buffer. Samples were resolved on a 4–20% gradient SDS-PAGE gel, transferred onto PVDF membranes and immunodetected using anti-mouse osteopontin (R and D Systems, Minneapolis, MN, USA) and anti-phosphoserine (Invitrogen, Cat# 61–

8100, Carlsbad, CA, USA) antibodies. Two technical replicates were performed for *Fgf23⁻/⁻* mice and corresponding WT littermates experiments while four technical replicates were performed for the Hyp and corresponding WT littermates experiments.

## In-vitro phosphorylation and dephosphorylation of whole mouse bone
### Sample preparation
Previously frozen femora were dissected from twenty-eight six-month-old male Opn KO (n = 14) and C57BL/6NCrl wild-type (WT, n = 14) mice. The sample size reflects the number of independent biological replicates and were based on results from previous pilot studies and publications from the laboratory (*Sroga and Vashishth, 2016*; *Poundarik et al., 2012*). Bones were cleaned of soft tissue and femoral head and condyle removed for experimental uniformed treatment throughout the bone. A notch was created on the anterior side in the mid-shaft of all samples using a slow speed diamond blade saw and sharpened using a razor blade (IsoMet Low Speed Saw, Buehler) This method produces a sharp notch with a root radius of ~10 µm (*Ritchie et al., 2008*). The crack length is defined in terms of the half crack angle and fracture toughness testing is accurate for half crack angles between 0–110 degrees (*Ritchie et al., 2008*). A specimen was considered an outlier and removed if crack angles were larger than two standard deviations from the mean, and if notches were off-centered or extended greater than 1/3 of the cortex. Consistent with physiological loading, the anterior side was chosen so that the notch experiences tension during bending test. The notch represents a pre-existing crack that will initiate and propagate into a large-scale catastrophic fracture. The bones were then rinsed with 1 x phosphate buffered saline (PBS) and stored in saline soaked gauze at −80°C until use.

### In-vitro phosphorylation and dephosphorylation
One limb of each animal (left or right) was randomly selected for phosphorylation and the contralateral limb served as the non-phosphorylated control [Opn KO (n = 7) and WT (n = 7)]. Phosphorylation was conducted by incubating the samples for 48 hr at 30°C with casein kinase-II (CK2) and the reaction buffer (New England BioLabs, Ipswich, MA). Adenosine triphosphate (ATP) (2 mM) was added to the buffer as the phosphoryl donor for CK2. The incubating solution also contained protease and phosphatase inhibitors (final concentration 2 x, Pierce Biotechnology, Rockford, IL), and antibiotics [ampicillin (100 µg/µL) and kanamycin (50 µg/µL)]. ATP, CK2, and antibiotics were also added second time to the reaction at the 20 hr of incubation. The non-phosphorylated samples (i.e. controls) were placed in a similar solution without added enzymes for the same time period and temperature.

In a different set of animals, one limb was randomly selected for de-phosphorylation and the contralateral limb served as the nondephosphorylated control [Opn KO (n = 7) and WT (n = 7)]. Dephosphorylation was conducted by incubating the samples for 48 hr at 37°C with calf intestinal alkaline phosphatase (CIP) and the CIP reaction buffer (New England BioLabs, Ipswich, MA). In pilot and published studies (*Sroga and Vashishth, 2016*) we did not observe increase in either phosphorylation or dephosphorylation of bone samples after 48 hr. The incubation solution also contained protease inhibitor and antibiotics as previously described. CIP enzyme was also added second time to the reaction at the 20 hr of incubation. The non-dephosphorylated samples (i.e. controls) were placed in a similar solution without added enzymes.

### Mechanical testing
All femora were scanned using micro-computed tomography (µCT) at 70 kVp, 114 mA, 200 ms integration time and at high resolution 10.5 µm voxel size (vivaCT 40, Scanco Medical AG, Bassersdorf, Switzerland) for measuring bone geometry. Following in-vitro phosphorylation and dephosphorylation treatment, samples were loaded in three-point bending until failure at a loading rate of 0.001 mm/s (Elf Enduratec 3200). The resulting load displacement curve was used to calculate a single-valued fracture toughness $K_c$ at maximum load for each sample (*Ritchie et al., 2008*). Toughness measured here is dependent on the material reflecting the changes due to phosphorylation or dephosphorylation.

## Protein extraction, quantification, and phosphoprotein detection

After mechanical testing, all bones were defatted, lyophilized (freeze-dried), and weighed (approximately 20–40 mg). Samples were then placed in eppendorf tubes with 600 µL of extraction buffer consisting of 0.05 M EDTA, 4 M guanidine chloride, and 30 mM Tris-HCl. The bones were subsequently homogenized, centrifuged, and the supernatant collected (Omni Inc, Kennesaw, GA). The supernatant from each sample was placed into a micro-dialysis vial and underwent simultaneous protein isolation and demineralization over two days at 4°C, pH 7.4, against several changes of 1 x PBS and 5 mM EDTA.

The amount of protein in the samples was quantified using the Coomassie Plus (Bradford) Assay. The measurement of phosphorylated proteins was done using the pIMAGO-biotin Phosphoprotein Detection assay kit (*Sroga and Vashishth, 2016*) (Tymora Analytical, West Lafayette, IN). Samples were tested in triplicates for each assay. Briefly, protein mixtures were bound to the wells by overnight incubation at 4°C. After a series of washing and blocking, the wells were incubated with pIMAGO reagent for attachment of the nanopolymer to phosphate groups on proteins. The wells were washed again, incubated with avidin-HRP followed by the provided colorimetric-based detection system. The absorbance was read at 415 nm using a micro-plate reader (Infinite M200, Tecan). The amount of global protein phosphorylation was calculated as absorbance/ng of protein. Assays for protein concentration and phosphoprotein detection were ran in triplicates.

## Data analysis for global phosphorylation

The mean and standard deviation were calculated for total protein phosphorylation amount and fracture toughness. Paired samples t-test was used to compare differences between the groups (WT-*phosphorylated vs.* WT-*nonphosphorylated;* Opn KO-*phosphorylated vs.* Opn KO -*non phosphorylated*). Because phosphorylation modifies the organic matrix including OPN, we compared the change in fracture toughness caused by phosphorylation of the organic matrix with (WT-treated _minus_ WT-control, delta-WT) and without osteopontin (Opn KO-treated _minus_ Opn KO -control, delta- Opn KO) by independent samples t-test. The same analysis was done for dephosphorylated samples and nondephosphorylated controls conducted on separate animals. All analyses were conducted using IBM SPSS 21 and two-tailed significance threshold set at 0.05 for both paired and independent samples t-test.

## Atomic force microscopy – force spectroscopy studies

### Chemicals

All chemicals were purchased from Sigma-Aldrich (Sigma-Aldrich Company Ltd., Gillingham, Dorset, UK) unless otherwise stated.

### Preparation and characterization of hydroxyapatite (HA) powder

Synthetic HA was produced for the functionalization of the AFM cantilever in order to simulate the mineralized fiber – NCP – mineralized fiber interaction. The preparation of the synthetic HA crystals was performed by the simultaneous addition of 250 mL aqueous solution of $H_3PO_4$ (0.3 M) and 250 mL aqueous solution of $CaCl_2 \cdot 2H_2O$ (0.5 M) to 500 mL ultrapure boiling water. To avoid temperature fluctuation, both reactants were added at a rate of approximately 2.5 mL per minute under continuous stirring. Prior and during the addition of the reactants, nitrogen gas was bubbled through the solution in order to remove the dissolved $CO_2$. At all times, the pH was kept between 9.0 and 10.0 by the addition of concentrated $NH_4OH$ solution. Upon the completion of the addition, the solution was kept under stirring for 24 hr at 80°C before cooling to room temperature. To retrieve the HA crystals, the suspension was filtered through a 0.22 µm membrane filters (Whatman, Maidstone England). Finally, the crystals were dried and 'matured' at 150°C for 24 hr and stored in a desiccator. The end product was characterized by means of X-Ray Diffraction (XRD), Fourier Transform Infrared Spectroscopy (FTIR) and Scanning Electron Microscopy equipped with Energy-dispersive X-ray analyser (SEM/EDX; Zeiss Supra 35VP). XRD analysis was performed using a standard powder diffractometer (Siemens D8) with Ni-filtered $CuKa_1$ radiation ($\lambda$ = 0.154059 nm) and the acquired diffraction spectra were matched against JCPDS reference data using the EVA XRD software. The FTIR spectra were acquired using an Excalibur spectrophotometer (Digilab, Japan) at a resolution of 2 $cm^{-1}$ using the KBr pellet method.

## Preparation and characterization of hydroxyapatite (HA) surfaces

HA surfaces were prepared through the 'maturation' of CaP cements in Ringer solution as described previously (*Knychala et al., 2013*; *Andriotis et al., 2010*). In brief, the cements were made by mixing alpha-tricalcium phosphate (a-TCP) powder with 4.0 % w/v disodium hydrogen phosphate ($Na_2HPO_4$) solution at liquid (mL)/powder (g) ratio of 0.32, homogenized by a spatula for 1 min in agate mortar and then spread carefully on Silastic M RTV Silicone Rubber moulds. The specimens were kept in 100% humidity for 12 hr and then placed in 60 mL of Ringer's solution at 37°C for 7 days to harden. During the maturation period, the a-TCP is transformed into calcium-deficient HA following the hydrolysis of the a-TCP according to the reaction $3Ca_3(PO_4)_2 + H_2O \rightarrow Ca_9(HPO_4)$ $(PO_4)_5OH$ (*Ginebra et al., 2004*).

## OPN purification and dephosphorylation

In bovine milk, OPN is subjected to proteolytic processing by proteinases such as thrombin (*Grassinger et al., 2009*), plasmin, cathepsin D or matrix metalloproteinases (*Christensen et al., 2010*). In this work isolated OPN from bovine milk as essentially described in *Sørensen and Petersen, 1993*. The principal components are N-terminal OPN fragments ending between residues 145 and 153 of the mature protein as well as the mature full-length protein (*Christensen and Sørensen, 2014*). After isolation, OPN was stored in a desiccator at room temperature until use. Dephosphorylated milk OPN was prepared as described in *Boskey et al., 2012*. Briefly, OPN was incubated with bovine alkaline phosphatase (ALP) (20 mU ALP/μg protein) in 10 mM ammonium bicarbonate (pH 8.5) overnight at 37°C and subsequently analyzed by MALDI-TOF MS to verify dephosphorylation.

## Buffer solutions

The buffers used were the same as previous studies (*Katayama et al., 1998*; *Fantner et al., 2005*; *Lai et al., 2014*). More specifically, Na Buffer (150 mM NaCl, 10 mM HEPES), Ca Buffer (40 mM $CaCl_2$, 110 mM NaCl, 10 mM HEPES), and ultra pure water ($H_2O$). Each solution was divided into separate vials and ph adjusted for each experiment using either HCl or concentrated NaOH solution.

## Adsorption of OPN on model surfaces

The lyophilized OPN was dissolved in ultrapure water (concentration 2 μg/μL) and absorption of OPN film on the model surfaces (HA or mica) was accomplished using the 'drying droplet' method. During this process, a small drop (4 μL) of OPN solution was deposited onto a freshly cleaned and dried HA or mica surface which was previously glued on the bottom of the fluid cell using 5 minute-setting epoxy. The droplet was then left to dry inside the AFM hood forming a thin protein film on the model surface, and then rehydrated with the appropriate solution.

## AFM cantilevers for force spectroscopy measurements

One aggregate of synthetic HA crystals was glued to a tipless monolithic silicon AFM probe (AIO-TL, Budget Sensors) using epoxy glue (Araldixe, Huntsman, The Woodlands, Texas, USA). For this, a few micrograms of the synthetic HA crystals were added in 5 mL of ethanol and stirred vigorously to produce a dispersion. At this stage, 500 μL of this dispersion were deposited onto a glass slide and left to dry. A droplet of epoxy was placed by the side of the dry crystals and the glass slide was placed into the AFM. The AFM probe was then engaged carefully onto the epoxy, pulled back, and engaged again on the aggregate of choice. After two minutes in contact, the probe was withdrawn and left in the AFM for an additional 30 min to ensure complete setting of the epoxy. An example of the end result is presented in *Figure 6—figure supplement 1*.

## Force spectroscopy experiments

Force spectroscopy measurements of the adhesive properties of the OPN film under various ionic environments were conducted by means of an atomic force microscope (MFP3D, Asylum Research, Santa Barbara, CA, USA) using an open fluid cell setup. Following *Fantner et al., 2005* protocol, all experiments were performed subsequently and at the same location. Exchange of solution, for altering the ionic environment and the pH, was carried using a syringe-pump inlet/outlet system without moving the head. In each environment, 50–80 pulls were collected and analyzed using a custom

made Matlab script (version 7.10.0.4999, The MathWorks Inc, Natick, Massachusetts, USA). For each force curve, the cantilever was positioned 3 μm away of the surface, driven in full contact with it, and after a dwell time of 10 s was retracted back to the starting position. During these cycles, the approach and retraction velocities were set to 2.0 μm/sec and 5.0 μm/sec, respectively. Full contact was defined as the tip-sample repulsive force reaching a threshold value of 15 nN. The spring constant, k, of the cantilever probe was measured prior to the functionalization using the thermal noise method (*Ritchie et al., 2008*), and followed by the Inverse Optical Lever Sensitivity (InvOLS) of the system. The later was determined by acquiring ten (10) force curves on a nominally infinitely stiff surface (*i.e.* the glass slide). A line was then fitted on the loading part of each force curve and the slope of the fitted line was used as the InvOLS. The mean InvOLS value of all ten curves was then used as the InvOLS of the cantilever. In the case of the HA-functionalized cantilevers the spring constant was reassessed using the thermal method post-functionalization and the resulting value was used for the analysis. Force spectroscopy measurements of phosphorylated/dephosphorylated OPN on mica surfaces were conducted using Olympus BL-RC150VB-C1 Bio-levers (Olympus Optical Co., Ltd., Tokyo, Japan); spring constant 6 pN/nm (0.006 N/m), while stiffer (c. 0.18 N/m) cantilevers were used for the HA experiments. Maximum force from force spectroscopy experiments are reported in *Supplementary files 3* and *4*.

## Data processing and analysis

All force curves were exported in ASCII (plain text files) and processed in Matlab. Each force curve was split into its approaching and retraction parts (*Figure 6—figure supplement 2*). Energy dissipation was defined as the area enclosed by the retraction curve and the X-axis from the point of contact (X = 0, Y = 0) to the 'Pulling Length'; where the latter was defined as the length from contact to the maximum distance at which the adhesion is smaller than the 1.0% of the Maximum Force (maximum adhesion force registered during retraction, i.e the Y-minimum of the retraction curve). Statistical analysis was performed in Origin (OriginPro version 9.0.0; OriginLab Corporation, Northampton, MA, USA). The normality of the distributions was assessed by means of a Kolmogorov-Smirnov test. Differences in Energy between the different environments were assessed by means of two-sided unpaired Student's t-test (significance threshold p=0.05).

# Acknowledgements

This work was supported by the following funding agencies: The National Institutes of Health AR 49635, Doctoral Prize Fellowship from Engineering and Physical Sciences Research Council (EPSRC), UK, the University of Southampton, UK, and the Canadian Institutes of Health Research. MDM is a member of the FRQS Network for Oral and Bone Health Research, and he holds the Canada Research Chair in Biomineralization as part of the Canada Research Chairs program which contributed to the funding of this work. We thank Beate Lanske for the generous dontation of the FGF23 knockout mice and to the Center for Biotechnology and Interdisciplinary Studies Imaging Core at Rensselaer Polytechnic Institute for providing access to micro-computed tomography.

# Additional information

### Competing interests

Marc D McKee: MDM is a member of the FRQS Network for Oral and Bone Health Research, and he holds the Canada Research Chair in Biomineralization as part of the Canada Research Chairs program which contributed to the funding of this work. The other authors declare that no competing interests exist.

### Funding

| Funder | Grant reference number | Author |
| --- | --- | --- |
| National Institutes of Health | AR 49635 | Stacyann Bailey<br>Grazyna E Sroga<br>Zehai Wang |

| | | Deepak Vashishth |
| --- | --- | --- |
| Canadian Institutes of Health Research | | Betty Hoac<br>Marc D McKee |
| University of Southampton | Doctoral Prize Fellowship | Orestis L Katsamenis |
| Canada Research Chairs | | Marc D McKee |
| EPSRC | Doctoral Prize Fellowship | Orestis L Katsamenis |

The funders had no role in study design, data collection and interpretation, or the decision to submit the work for publication.

## Author contributions

Stacyann Bailey, Data curation, Formal analysis, Validation, Investigation, Methodology, Writing - original draft, Writing - review and editing; Grazyna E Sroga, Conceptualization, Supervision, Validation, Methodology, Writing - original draft, Writing - review and editing; Betty Hoac, Orestis L Katsamenis, Data curation, Software, Formal analysis, Validation, Investigation, Methodology, Writing - original draft, Writing - review and editing; Zehai Wang, Software, Formal analysis, Validation, Investigation, Methodology, Writing - original draft; Nikolaos Bouropoulos, Resources, Methodology, Writing - review and editing; Marc D McKee, Conceptualization, Resources, Supervision, Funding acquisition, Investigation, Methodology, Writing - review and editing; Esben S Sørensen, Resources, Investigation, Methodology, Writing - review and editing; Philipp J Thurner, Conceptualization, Resources, Software, Supervision, Funding acquisition, Methodology, Writing - original draft, Writing - review and editing; Deepak Vashishth, Conceptualization, Resources, Supervision, Funding acquisition, Methodology, Writing - review and editing

## Author ORCIDs

Stacyann Bailey (iD) https://orcid.org/0000-0001-9013-2469
Esben S Sørensen (iD) https://orcid.org/0000-0002-7050-3354

## Ethics

Animal experimentation: This study was performed in strict accordance with the recommendations in the Guide for the Care and Use of Laboratory Animals of the National Institutes of Health. All of the animals were handled according to approved institutional animal care and use committee (IACUC) protocols (VAS-001-14) of Rensselaer Polytechnic Institute.

## Decision letter and Author response

Decision letter https://doi.org/10.7554/eLife.58184.sa1
Author response https://doi.org/10.7554/eLife.58184.sa2

# Additional files

## Supplementary files

• Supplementary file 1. Adhesive properties of native (phosphorylated) and dephosphorylated OPN film on mica.

• Supplementary file 2. Adhesive properties of native (phosphorylated) OPN film on HA.

• Supplementary file 3. Mean maximum force of native (phosphorylated) OPN film on HA.

• Supplementary file 4. Mean maximum force of native (phosphorylated) and dephosphorylated OPN film on mica.

• Transparent reporting form

## Data availability

All data generated or analyzed during this study are included in the manuscript and supporting files.

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
