## [Decision Letter]

**Acceptance summary:**

Increasing attention in the bone field is focused on understanding the nuances of the composition of extracellular matrices and how post translational modifications of constituent proteins alters cellular interactions, tissue integrity and function. In an elegant series of experiments, the authors show that the phosphorylation status of osteopontin can be modulated, altering its energy dissipation properties thereby promoting fracture toughness, but that there is a ceiling to this effect. The mechanisms by which phosphorylation alters facture toughness are explored in detail and these results may be informative for understanding the properties of the extracellular matrix in other connective tissues.

**Decision letter after peer review:**

Thank you for submitting your article "The Role of Extracellular Matrix Phosphorylation on Energy Dissipation in Bone" for consideration by *eLife*. Your article has been reviewed Clifford Rosen as the Senior Editor, a Reviewing Editor, and three reviewers. The following individual involved in review of your submission has agreed to reveal their identity: Virginia Ferguson (Reviewer #1).

The reviewers have discussed the reviews with one another and the Reviewing Editor has drafted this decision to help you prepare a revised submission.

We would like to draw your attention to changes in our revision policy that we have made in response to COVID-19 (https://elifesciences.org/articles/57162). Specifically, when editors judge that a submitted work as a whole belongs in *eLife* but that some conclusions require a modest amount of additional new data and or revision, as they do with your paper, we are asking that the manuscript be revised to either limit claims to those supported by data in hand, or to explicitly state that the relevant conclusions require additional supporting data.

Summary:

There is a great deal of interest in the field in determining how the non-collagenous aspects of the bone matrix contribute to the overall resistance of bone tissue to fracture. In this manuscript, the authors seek to investigate the importance of phosphorylation status of osteopontin to bone toughness and how this non-collagenous extracellular matrix protein impacts fracture toughness. Of particular interest in this paper is how the contribution of osteopontin phosphorylation to whole bone toughness is explored at the nano-scale, protein-mineral interactions level. They determine that the phosphorylation of OPN increases electrostatic interactions and cation-mediated crosslinks, allowing for increased energy dissipation under loading. Overall, this was considered to be a well written and interesting paper comprised of well thought out experiments.

During the review many requests for additional information and clarification were made by the reviewers but few experiments. The reviewers wondered if alternate mechanisms could be competing with OPN towards affecting fracture toughness. In other instances, it was wondered if alternate explanations might not explain these data. These concerns are summarized below.

Essential revisions:

1) The limitations of this study were glossed over in terms of the limits of the OPN^-/-^ mouse model (e.g., what else is changing in the OPN deficient bone during development that could influence experimental outcomes?), and what sources of error may exist and be generated from specific methods (e.g., AFM testing or the intrinsic variability of notch fracture testing of whole mouse bones).

2) Could there be additional interactions that were potentially overlooked where phosphorylation status changes the way that OPN interacts with other matrix molecules in bone? What about the pH effects on other proteins, GAGs, etc. in bone – could these have complicated or contributed to the results that were observed in this manuscript in any meaningful way? The sole focus of this manuscript on OPN is beneficial, but it also excludes many other concomitant effects of phosphorylation/dephosphorylation and pH that could compound the experimental outcomes that were observed. The authors are asked to briefly justify (and consider including in their manuscript) a justification of why these effects on other matrix molecules were not the dominant outcome – in other words that OPN is the sole or predominant contributor to the endpoint measurements of interest (e.g., fracture toughness).

3) Figure 4, Figure 5 include bars for statistical comparisons (marked with *) that are sometimes not located in a manner to compare two groups. It is thus difficult to see which groups are intended to be marked in a comparison. Figure 5 also seems to have lost the -/- for OPN^-/-^ group within the figure legend, and Figure 4B is missing "C" in the "Change in.…" title. To provide readers a transparent interpretation of the results, consider overlaying dot plots on the bar charts. That is, show the actual data points. This will also convey the sample size more clearly.

4) Data from control mice in Figure 4 and Figure 5 don't seem to match. What could this mean when the differences in means are comparable to those observed in some of your experimental comparisons? Does this cast doubt on the overall results of fracture toughness testing? How can we believe the results you present if the WT groups are not producing repeatable results?

5) Why was 48 hours chosen for phosphorylation and dephosphorylation? Is the (de)phosphorylation achieved with this protocol physiologically-relevant?

6) The position and mass of the HA cluster on the cantilever would be expected to affect the spring constant of the cantilever. How was this accounted for? It is only mentioned that the cantilevers were calibrated before functionalization.

7) For AFM testing, how was the effect of different shaped crystal clusters on electrostatic interaction forces accounted for? Numbers of replicates/group for AFM testing and some other assays were unclear. This is essentially a contact mechanics question, since the surface area of contact could be quite different for different clusters. More detail about the normalization procedure could help here (was the normalization for the same functionalized HA geometry?). How was the full contact threshold force decided? For experiments with changing buffers, was the order of buffer change randomized? Otherwise, buffer effects may be confounded with washing away part of the OPN or the crystal.

8) Did mineralization or collagen crosslinking change with phosphorylation or dephosphorylation treatments of femurs? The authors state that "loss of OPN resulted in reduced fracture toughness without accompanied changes in calcium variability", but it is not clear if it is assumed that mineralization did not change with (de)phosphorylation processes.

9) Statistics: In the case of a significant interaction, which the authors have found, simple effects can be tested, but family-wise error should be considered and accounted for to correct for the greater likelihood of Type I error. It’s not correct to t-test first and then use ANOVA to look for an interaction. It is better to fit your data to a two-factor ANOVA, and in the case of an interaction, perform post-hoc simple effects testing that is corrected for these multiple comparisons. Further, details ensuring model fit (normality and homoscedascity of residuals) would be appreciated. It is suggested to present data as a mean instead of an average. "Average" can be defined as mean, median, mode… etc.

10) The sentence “Treatment affected fracture toughness of WT bone…” is confusing. There is a significant interaction, and that p-value should be reported, but instead, p-values are reported for phosphorylation and dephosphorylation.

11) For quantification of phosphorylation in Hyp and FGF23^-/-^ mice, the mice were very young (6 weeks old). The mice were much older (6 months) for fracture toughness testing. Phosphorylation changes with aging; how much less phosphorylation would these mouse models have at 6-mo of age, and how would this relate to the WT mice?

12) The fracture toughness of bone originates from nanoscale energy dissipation at the protein-mineral interface. While this toughening mechanism is important, it is not the only important toughening mechanism in bone. Toughening mechanisms span length-scales and involve mineral, collagen, and their interactions. A slight modification to language to contextualize this toughening mechanism as an important one of several would be helpful.

13) The authors' finding that there is a “sweet spot” of phosphorylation was considered to be very interesting. However, this concept of sweet spot confused the reviewers in the manner that it was presented. The authors did not provide a fracture toughness vs. phosphorylation relationship. That is, by continuing the increase in phosphorylation of WT bone, does the improvement in fracture toughness stop and decline? With the loss of OPN, phosphorylation had a negative effect. This was interpreted to mean that phosphorylation levels returned to normal, but really, the finding implicates the phosphorylation of OPN as being required to improve toughness. It is not clear what subset of phosphorylation values within the studied range of phosphorylation here is relevant in disease. The “sweet spot” mechanism likely needs to be removed or qualified. Either way, its description is rather difficult to follow.

14) The in vitro (de)phosphorylation had some variability. Did amount of (de)phosphorylation correlate (within groups) with fracture toughness?

15) The authors postulate that TG2 and phosphorylation may separately act as toughening mechanism. This is supported by FGF23 and Hyp mice having increased TG2, which should toughen bone, but also phosphorylation. But what if TG2 also has a nonlinear relationship with bone mechanics? The reviewers were concerned that there was not enough evidence in the paper to say that TG2 and phosphorylation are independent, but felt this is a very interesting discussion point. Absolutes should be tempered here.

16) A reviewer asked if one could over-phosphorylate the bovine OPN to potentially observe the quadratic relationship with energy dissipation for confirmation of the relationship observed for whole-bone testing? Some discussion on this point would be appreciated.

[Editors' note: further revisions were suggested prior to acceptance, as described below.]

Thank you for resubmitting your work entitled "The Role of Extracellular Matrix Phosphorylation on Energy Dissipation in Bone" for further consideration by *eLife*. Your revised article has been evaluated by Clifford Rosen (Senior Editor) and a Reviewing Editor.

The manuscript has been improved but there are some remaining issues that need to be addressed before formal acceptance, as outlined below. Please note that these requested clarifications were agreed as necessary by all reviewers during the post review correspondence.

Reviewer #1:

This manuscript is well written and is on a topic that is compelling and important to the field. The authors have given sufficient attention to the concerns expressed by the reviewers, including myself. I feel satisfied that the responses were diligent and thorough and the manuscript is improved as a result.

Upon revisiting Figure 8A, I agree that a quadratic fit is not necessarily the best choice and that an exponential function should be considered. Concerningly, if the very most highly phosphorylated point is excluded, the data look very much like a plateau. This would challenge the “sweet spot” idea as applied to just WT bones. As reviewer 3 notes it “seems like the study showed that phosphorylation promotes fracture toughness (to a point) if OPN is present”. I agree with this interpretation based on the presented data, which is consistent with more of a asymptote than a sweet spot in Figure 8A.

Reviewer 3 asks “why doesn't the increase in fiber network modulus and strength continue to increase in phosphorylation”? I read this differently, with the interpretation that modulus and strength could indeed increase with phosphorylation but to a detrimental degree that decreases strain. Clarification is required as the reader will likely be confused on this point as well.

It would be helpful to show all data (not just four means) for Figure 8B and to indicate which data correspond with each group. This would alleviate confusion surrounding this figure.

The authors should clarify in the Abstract that the phosphorylation and dephosphorylation experiments were conducted ex vivo.

Reviewer 3 comments that the abstract only mentions one mechanism of the three in the discussion. My bigger issue is that mechanisms 1 and 3 are co-dependent as presented (they are described as "three different, co-existing mechanisms". The authors argue that mineral-matrix interactions are facilitated by OPN phosphorylation, but this would be the real mechanism and the “sweet spot” or “benefit to a point” would be the relationship of this mechanism to dose.

The authors should add a little bit of clarifying statistical information (1 vs 2 tailed testing, critical α). These are stated for the AFM experiments but not for the fracture toughness and phosphorylation experiments.

Reviewer #2:

The revised manuscript is considerably improved. The detailed Response to reviewers is appreciated, as well as the updates to figures and text. The revised Figure 8 is particularly appreciated, notwithstanding the comments made by other reviewers.

I agree that their findings are exiting and that the explanations need to be improved so that the readers do not need to make inferences that may be off track.

Reviewer #3:

The study is intriguing because it shows how phosphorylation promotes fracture toughness of bone if osteopontin is present. However, this reviewer still struggles with the authors' interpretation of the mechanisms by which phosphorylation does this (they propose 3). A number of questions come up after reading the Abstract and Discussion. I like the concept that modulus and strength increases with phosphorylation but at a cost of decreasing strain after some threshold. The authors just need to do a better job describing each mechanism and how they are inter-related so that readers don't have to interpret their interpretation.

How did the authors decide on fitting a quadratic equation to fracture toughness vs. phosphorylation levels of WT bone? To this reviewer, it looks like fracture toughness plateaued after 1.5 in Figure 8A. What about fitting an exponential function, exp(1/phos)?

Discussion: "However, further increase in the degree of most favorable interaction (i.e. larger than the "sweet spot" level) can cause a rapid decrease in ultimate strain, and thus, energy dissipation." What is the physical mechanism that supports this assertion? In other words, why doesn't the increase fiber network modulus and strength continue to increase with an increase in phosphorylation?

Another thing that is challenging to the interpretation of the results is that fact OPN bridges mineral crystals, presumably more bridges with more phosphorylation. Correct? This is the first mechanism described. In the OPN^-/-^ bone, this mechanism is gone. Is the relationship between fracture toughness of OPN^-/-^ bone and phosphorylation levels (from the 2 enzyme treatments) non-linear? Sorry, Figure 8B is not too convincing with only 4 data points.

---

## [Author Response]

Essential revisions:1) The limitations of this study were glossed over in terms of the limits of the OPN ^-/-^ mouse model (e.g., what else is changing in the OPN deficient bone during development that could influence experimental outcomes?), and what sources of error may exist and be generated from specific methods (e.g., AFM testing or the intrinsic variability of notch fracture testing of whole mouse bones).

The limitations of the study are now reported in the Discussion.

2) Could there be additional interactions that were potentially overlooked where phosphorylation status changes the way that OPN interacts with other matrix molecules in bone? What about the pH effects on other proteins, GAGs, etc. in bone – could these have complicated or contributed to the results that were observed in this manuscript in any meaningful way? The sole focus of this manuscript on OPN is beneficial, but it also excludes many other concomitant effects of phosphorylation/dephosphorylation and pH that could compound the experimental outcomes that were observed. The authors are asked to briefly justify (and consider including in their manuscript) a justification of why these effects on other matrix molecules were not the dominant outcome – in other words that OPN is the sole or predominant contributor to the endpoint measurements of interest (e.g., fracture toughness).

Our experiments were conducted at pH 7.4 under physiological conditions, and any instability in pH would affect proteins in both WT and *Opn^-/-^* groups. By using *Opn^-/-^* mice we can determine the contribution of all other matrix proteins in the absence of OPN. Likewise, by comparing the change in fracture toughness caused by phosphorylation or dephosphorylation of the organic matrix with osteopontin (WT-treated minus WT-control, δ-WT) and without osteopontin (*Opn^-/-^* -treated minus *Opn^-/-^* -control, δ-*Opn*^-/-^), we can isolate the contribution from OPN alone.

OPN in bone resides at its surfaces (including lining the lacuno-canalicular system) in the thin structure known as the *lamina limitans* (McKee and Nanci, 1996), and throughout bulk bone, so alterations in vivo may affect many processes where phosphorylation is important. These would seem to be predominantly involved in mineral-binding and possibly modifying TG2 crosslinking sites, and in cell signaling that may affect mechanosensation and bone remodeling. Dephosphorylation of OPN by the enzymatic activity of TRAP also affects osteoclast integrin binding in vitro, and thus altered bone remodeling over time affecting bone's mechanical properties. In addition to its interaction with mineral, the phosphorylation status of OPN is likely to influence its interaction with collagen and osteocalcin. Reflecting this, new text has been added to the Discussion section.

3) Figure 4, Figure 5 include bars for statistical comparisons (marked with *) that are sometimes not located in a manner to compare two groups. It is thus difficult to see which groups are intended to be marked in a comparison. Figure 5 also seems to have lost the -/- for OPN ^-/-^ group within the figure legend, and Figure 4B is missing "C" in the "Change in.…" title. To provide readers a transparent interpretation of the results, consider overlaying dot plots on the bar charts. That is, show the actual data points. This will also convey the sample size more clearly.

As suggested, these figures have been corrected. Thank you.

4) Data from control mice in Figure 4 and Figure 5 don't seem to match. What could this mean when the differences in means are comparable to those observed in some of your experimental comparisons? Does this cast doubt on the overall results of fracture toughness testing? How can we believe the results you present if the WT groups are not producing repeatable results?

There is inherent variability in fracture toughness testing despite the fact that the maximum load method for measuring fracture toughness of mice bone used in this study has the least variability compared to other methods (Ritchie et al., 2008). We and others have shown with a larger sample size that *Opn*^-/-^ mice have lower fracture toughness compared to WT mice (Thurner et al., 2010; Poundarik et al., 2012). This study was not powered to detect differences between controls of different genotypes, but rather the experimental design was to examine differences between paired groups i.e. WT-dephosphorylated vs WT-controls. The data in Figure 4 and Figure 5 are from a different set of control bones and fracture toughness values vary across bones from the same batch of mice due to inherent differences between animals, and due to variations in any mechanical testing method including fracture toughness testing. In this case, we have minimized variations between animals by conducting pairwise comparison (where one bone is kept as control and the other modified in vitro under carefully controlled conditions, thus only one factor i.e. phosphorylation/dephosphosphorylation is changed). This has been noted in the Discussion section.

5) Why was 48 hours chosen for phosphorylation and dephosphorylation? Is the (de)phosphorylation achieved with this protocol physiologically-relevant?

Our in vitro studies were performed using enzymes which require physiological conditions for the reaction to be successful. The physiological reaction time for a given enzyme, both in vivo and in vitro, is its lifetime. All enzymes stop to function (i.e. “die”) due to, for example, enzyme unfolding, damage to the active center, adsorption on the tube wall, etc. In pilot and published studies (Sroga and Vashishth, 2016) we did not observe an increase in either phosphorylation or dephosphorylation of bone samples after 48 hours. This has been noted in methods section in vitro Phosphorylation and Dephosphorylation.

6) The position and mass of the HA cluster on the cantilever would be expected to affect the spring constant of the cantilever. How was this accounted for? It is only mentioned that the cantilevers were calibrated before functionalization.

Thank you for raising this point. The text has been modified in subsection “Force spectroscopy experiments” to include the requested information. In the case of the HA-functionalized cantilevers, the calibration of the cantilever’s spring constant, done prior to functionalization, was used to calibrate the Inverse Optical Lever Sensitivity (InvOLS), which required force curves to be acquired on a nominally infinitely stiff surface. (The Inverse Optical Lever Sensitivity (InvOLS) was determined by acquiring ten (10) force curves on a nominally infinitely stiff surface (i.e. the glass slide). A line was then fitted on the loading part of each force curve and the slope of the fitted line was used as the InvOLS. The mean InvOLS value of all ten curves was then used as the InvOLS of the cantilever.) The reason we avoided calibrating the InvOLS post-functionalization was to reduce the chance of mechanical damage to the functionalized end of the cantilever, which could affect both the adhesive (epoxy) and the integrity of the HA aggregate.

Following InvOLS calibration and cantilever functionalization, the spring constant was re-assessed using the thermal noise method and the new value was used for the analysis. However, we agree that the presence of a foreign body on the cantilever can influence its behavior. To ensure comparable results, care was taken so that the selected HA aggregate was always of similar shape and size. This was further confirmed post-calibration by assessing the amount of cantilever deflection. It is worth noting that small variations in spring constant calibration would mainly affect the reading of maximum force and, in this study, we are presenting and discussing the relative energy dissipation change between the OPN groups, which is primarily affected by the pulling Length.

7) For AFM testing, how was the effect of different shaped crystal clusters on electrostatic interaction forces accounted for? Numbers of replicates/group for AFM testing and some other assays were unclear. This is essentially a contact mechanics question, since the surface area of contact could be quite different for different clusters. More detail about the normalization procedure could help here (was the normalization for the same functionalized HA geometry?). How was the full contact threshold force decided? For experiments with changing buffers, was the order of buffer change randomized? Otherwise, buffer effects may be confounded with washing away part of the OPN or the crystal.

Our experiments were conducted in relative terms; thus, correcting for variations of crystal cluster shape was not applicable. We normalized our measurements to the dissipation levels in EDTA (for OPN on mica) and to dissipation levels in H_2_O (for OPN on HA). For each condition, we collected 50 < n < 60 force curves for the OPN on HA experiments and 80 < n < 100 for the OPN on mica experiment. Full contact was ensured by pressing the cantilever against the coated surface to the point where the force curve slope approached infinity, condition which was achieved for contact forces > 5 nN. The order of the buffer change was not randomized. It followed the order shown in Figure 6 (H_2_O, Na, Ca).

8) Did mineralization or collagen crosslinking change with phosphorylation or dephosphorylation treatments of femurs? The authors state that "loss of OPN resulted in reduced fracture toughness without accompanied changes in calcium variability", but it is not clear if it is assumed that mineralization did not change with (de)phosphorylation processes.

The lack of changes in calcium variability was reported in Thurner et al., 2010. Crosslinking between collagen remains unchanged. Dephosphorylation reverses the inhibitory effect of OPN on HA formation in vitro and thus mineralization may be affected. In this study, the experiments were conducted under physiological conditions with buffer solutions containing magnesium chloride, calcium, and EDTA to prevent any alterations in mineral. Thus, our conditions do not induce crystallization. This has been noted in the Discussion section.

9) Statistics: In the case of a significant interaction, which the authors have found, simple effects can be tested, but family-wise error should be considered and accounted for to correct for the greater likelihood of Type I error. It’s not correct to t-test first and then use ANOVA to look for an interaction. It is better to fit your data to a two-factor ANOVA, and in the case of an interaction, perform post-hoc simple effects testing that is corrected for these multiple comparisons. Further, details ensuring model fit (normality and homoscedascity of residuals) would be appreciated. It is suggested to present data as a mean instead of an average. "Average" can be defined as mean, median, mode… etc.

Kindly note that, as mentioned above and in the manuscript (Discussion section, Materials and methods secrtion), this study is motivated from results showing differences between *Opn^-/-^* and WT mouse bone, and the goal was to test whether modification of OPN by phosphorylation (and not all matrix proteins) affect the outcome. Thus, given that in vivo phosphorylation and dephosphorylation of WT and *Opn^-/-^* mice may cause changes other than in the level of phosphorylated OPN, we designed our experimental method where WT and *Opn^-/-^* mouse bones were paired and altered in vitro under carefully controlled conditions to modify a selected aspect. Pairwise comparisons where both limbs are from the same animal are valid (for example, comparing WT-phosphorylated vs WT-non-phosphorylated control where both limbs are from the same animal; likewise, comparing *Opn^-/-^* phosphorylated vs *Opn^-/-^* non-phosphorylated controls). Both Kolmogorov-Smirnov and Shapiro-Wilk Tests of Normality revealed that phosphorylation levels and fracture toughness were normally distributed (p>0.05). Next, to determine whether phosphorylation of osteopontin contributes to fracture toughness, we compared δ-WT (the change between WT-phosphorylated and WT-non-phosphorylated control), and δ-*Opn^-/-^* (the change between *Opn^-/-^* phosphorylated vs *Opn^-/-^* non-phosphorylated controls). By comparing δ-WT and δ-*Opn^-/-^* the difference detected would be attributed to the presence of osteopontin since all other proteins that were phosphorylated would be subtracted as background. Two-way ANOVA tests are not designed to evaluate the difference between WT control and WT phosphorylated and *Opn^-/-^* control and *Opn^-/-^* phosphorylated but between the mean values of these four groups and do not allow us to test whether phosphorylation of OPN causes a difference.

10) The sentence “Treatment affected fracture toughness of WT bone…” is confusing. There is a significant interaction, and that p-value should be reported, but instead, p-values are reported for phosphorylation and dephosphorylation.

Please see the response to the above comment where we include the rationale of not performing two-way ANOVA tests. To avoid any confusion; this sentence has been removed.

11) For quantification of phosphorylation in Hyp and FGF23^-/-^ mice, the mice were very young (6 weeks old). The mice were much older (6 months) for fracture toughness testing. Phosphorylation changes with aging; how much less phosphorylation would these mouse models have at 6-mo of age, and how would this relate to the WT mice?

This is a very interesting question. Our assay for the quantification of global protein phosphorylation was recently developed (Sroga and Vashishth, 2016) and to date no study has quantified the amount of phosphorylation in normal and diseased bone during development. We have previously reported that total phosphorylation of bone matrix proteins including OPN declines with age (Sroga and Vashishth, 2018). Despite our looking at only one mouse age so far, our results here are the first to show that global phosphorylation, as well as OPN phosphorylation, declines in osteomalacic *Hyp* and *Fgf23*^-/-^ mice, and in our view, this makes an important initial contribution that we chose to mention here. Quantification of the amount of phosphorylation in these models, and at different ages, will be done in future studies as an additional full-scale project. Fracture toughness testing on the 6-week old mice used here is very difficult to perform, and less reliable, but older mice will be incorporated into our future planned work on these mutant mice.

12) The fracture toughness of bone originates from nanoscale energy dissipation at the protein-mineral interface. While this toughening mechanism is important, it is not the only important toughening mechanism in bone. Toughening mechanisms span length-scales and involve mineral, collagen, and their interactions. A slight modification to language to contextualize this toughening mechanism as an important one of several would be helpful.

The statement has been clarified to “Fracture resistance of bone emerges from various mechanisms that exists at multiple length scales across bone hierarchy, which involves in part growth and packing of mineral foci into larger crossfibrillar aggregates such that the extracellular matrix becomes highly mineralized[…]Both the OPN-crosslinked protein networks and the OC-OPN complex presumably work together to control deformation and separation of mineralized collagen fibrils” please see Discussion section.

13) The authors' finding that there is a “sweet spot” of phosphorylation was considered to be very interesting. However, this concept of sweet spot confused the reviewers in the manner that it was presented. The authors did not provide a fracture toughness vs. phosphorylation relationship. That is, by continuing the increase in phosphorylation of WT bone, does the improvement in fracture toughness stop and decline? With the loss of OPN, phosphorylation had a negative effect. This was interpreted to mean that phosphorylation levels returned to normal, but really, the finding implicates the phosphorylation of OPN as being required to improve toughness. It is not clear what subset of phosphorylation values within the studied range of phosphorylation here is relevant in disease. The “sweet spot” mechanism likely needs to be removed or qualified. Either way, its description is rather difficult to follow.

Thank you for the suggestion. We have revised Figure 8 and provided a fracture toughness vs phosphorylation graph using the values from the study. By continuing the increase in phosphorylation of WT bone, fracture toughness does improve and then declines. This relationship is modeled by following equation y = 2.086+ 2.574x- 0.6415x^2^ (R^2^=0.49, F=7.72, p=0.00450).

14) The in vitro (de)phosphorylation had some variability. Did amount of (de)phosphorylation correlate (within groups) with fracture toughness?

Due to the variability, the amount of dephosphorylation did not correlate with fracture toughness. In addition, the correlation coefficient will detect a linear relationship, but our data shows that global phosphorylation levels in WT bone follows a non-linear relationship as shown in comment #12.

15) The authors postulate that TG2 and phosphorylation may separately act as toughening mechanism. This is supported by FGF23 and Hyp mice having increased TG2, which should toughen bone, but also phosphorylation. But what if TG2 also has a nonlinear relationship with bone mechanics? The reviewers were concerned that there was not enough evidence in the paper to say that TG2 and phosphorylation are independent, but felt this is a very interesting discussion point. Absolutes should be tempered here.

We agree that there is not enough evidence and have tempered down the independence of TG2 and phosphorylation. Please see the Discussion section.

16) A reviewer asked if one could over-phosphorylate the bovine OPN to potentially observe the quadratic relationship with energy dissipation for confirmation of the relationship observed for whole-bone testing? Some discussion on this point would be appreciated.

Thank you for this suggestion. We have added relevant discussion on this topic (please see the Discussion section). In particular, OPN is phosphorylated by the FAM20C kinase. The primary motif for phosphorylation is S/T-X-E/S(P)/D. Bovine milk OPN contains approximately 28 phosphorylation sites and all but a few residues in this motif are phosphorylated. If more sites should be phosphorylated, a kinase that phosphorylates serines/threonines in another motif should be used. However, in this context, we do not think that this approach would be successful as the non-phosphorylated serines in bovine OPN are not located in recognition sequences of any specific kinase. Therefore, milk OPN is already close to the over-phosphorylated OPN animal model. In hindsight, it may be possible to perform measurements on partially de-phosphorylated OPN and, in this sense, assess the whole curve shown. However, we note that several assays would be necessary to characterize the modified protein. Importantly, the consequence for mechanical properties of OPN networks is also influenced by the abundance and concentration of Ca^2+^ ions. As shown in the AFM experiments, situations that lead to high Ca^2+^ ion concentration result in shielding of the phosphorylated sites, charge reversal and repulsion. While the lack of experiments on OPN with varying phosphorylation-levels may be seen as a limitation, we nonetheless provide data points for the most extreme cases, and, different to the physiological system, control the concentration of Ca^2+^ ions. Taken together, our experiments do reproduce the effects seen in the animal models. We are planning to do further studies with variably and reproducibly phosphorylated OPN, as has recently been achieved in co-author Sorensen's lab.

[Editors' note: further revisions were suggested prior to acceptance, as described below.]

Reviewer #1:This manuscript is well written and is on a topic that is compelling and important to the field. The authors have given sufficient attention to the concerns expressed by the reviewers, including myself. I feel satisfied that the responses were diligent and thorough and the manuscript is improved as a result.Upon revisiting Figure 8A, I agree that a quadratic fit is not necessarily the best choice and that an exponential function should be considered. Concerningly, if the very most highly phosphorylated point is excluded, the data look very much like a plateau. This would challenge the “sweet spot” idea as applied to just WT bones. As reviewer 3 notes it 'seems like the study showed that phosphorylation promotes fracture toughness (to a point) if OPN is present. I agree with this interpretation based on the presented data, which is consistent with more of a asymptote than a sweet spot in Figure 8A.

Thank you for raising this point. Figure 8A has been revised with an exponential function which is also significant (p<0.001). Consistent with suggestion from reviewer #3, our data shows that fracture toughness increases with phosphorylation only when OPN is present. Figure 8B has been added to reinforce this point.

Reviewer 3 asks “why doesn't the increase in fiber network modulus and strength continue to increase in phosphorylation”? I read this differently, with the interpretation that modulus and strength could indeed increase with phosphorylation but to a detrimental degree that decreases strain. Clarification is required as the reader will likely be confused on this point as well.

Based on the revised Figure 8A, these previous statements have been removed and we provide the following information to explain the increase of fracture toughness with phosphorylation for wildtype samples (Discussion section):

“We observed a non-linear dose response relationship between the level of global matrix phosphorylation and bone fracture toughness in wildtype mice (Figure 8A). Phosphorylation explained ~36% of the variance in fracture toughness and this relationship was not observed in the absence of OPN (Figure 8B). Taken together, this data supports the previously mentioned mechanism involving increased interaction energy and sacrificial bone formation between OPN and HA as well as between OPN polymers. The AFM-FS studies show that adhesion is not only dependent on the charge of OPN and HA under a certain environment but also the availability of free Ca^2+^ ions”

It would be helpful to show all data (not just four means) for Figure 8B and to indicate which data correspond with each group. This would alleviate confusion surrounding this figure.

As described above in response #1 the previous Figure 8A and the related sweet spot hypothesis have been removed. Thank you for the suggestion, all the data points are now included in Figures 8A and 8B.

The authors should clarify in the Abstract that the phosphorylation and dephosphorylation experiments were conducted ex vivo.

This has been clarified. We now state that “Fracture toughness, a measure of bone’s mechanical competence, increased with *ex-vivo* phosphorylation of wildtype mouse bones and declined with *ex-vivo* dephosphorylation. In osteopontin-deficient mice, global matrix phosphorylation level was not associated with toughness.”

Reviewer 3 comments that the abstract only mentions one mechanism of the three in the discussion. My bigger issue is that mechanisms 1 and 3 are co-dependent as presented (they are described as "three different, co-existing mechanisms". The authors argue that mineral-matrix interactions are facilitated by OPN phosphorylation, but this would be the real mechanism and the “sweet spot” or “benefit to a point” would be the relationship of this mechanism to dose.

Thank you for the suggestion. We have now clarified that mineral-matrix interactions facilitated by OPN phosphorylation, is the mechanism for the non-linear exponential increase in toughness with phosphorylation to a point. The second mechanism explains how phosphorylation of bone matrix in the absence of Osteopontin affects fracture toughness. The explanation has been added in Discussion section and in the Abstract.

The authors should add a little bit of clarifying statistical information (1 vs 2 tailed testing, critical α). These are stated for the AFM experiments but not for the fracture toughness and phosphorylation experiments.

Statistical clarification for the phosphorylation and fracture toughness experiments are now included in subsection “Data analysis for global phosphorylation”. We now state “All analyses were conducted using IBM SPSS 21 and two-tailed significance threshold set at 0.05 for both paired and independent samples t-test.”

Reviewer #2:The revised manuscript is considerably improved. The detailed Response to reviewers is appreciated, as well as the updates to figures and text. The revised Figure 8 is particularly appreciated, notwithstanding the comments made by other reviewers.I agree that their findings are exiting and that the explanations need to be improved so that the readers do not need to make inferences that may be off track.

Thank you for the feedback. Figure 8 has been revised and the Discussion has been reorganized to limit inferences from the reader.

Reviewer #3:The study is intriguing because it shows how phosphorylation promotes fracture toughness of bone if osteopontin is present. However, this reviewer still struggles with the authors' interpretation of the mechanisms by which phosphorylation does this (they propose 3). A number of questions come up after reading the Abstract and Discussion. I like the concept that modulus and strength increases with phosphorylation but at a cost of decreasing strain after some threshold. The authors just need to do a better job describing each mechanism and how they are inter-related so that readers don't have to interpret their interpretation.How did the authors decide on fitting a quadratic equation to fracture toughness vs. phosphorylation levels of WT bone? To this reviewer, it looks like fracture toughness plateaued after 1.5 in Figure 8A. What about fitting an exponential function, exp(1/phos)?

Thank you for the suggestion. Kindly note that as mentioned above, an exponential function rather than a quadratic function is now used to fit the relationship between phosphorylation levels and fracture toughness of WT bone subjected to ex vivo phosphorylation/dephosphorylation (Figure 8A).

Discussion: "However, further increase in the degree of most favorable interaction (i.e. larger than the "sweet spot" level) can cause a rapid decrease in ultimate strain, and thus, energy dissipation." What is the physical mechanism that supports this assertion? In other words, why doesn't the increase fiber network modulus and strength continue to increase with an increase in phosphorylation?

It is possible that fiber realignment and altering the net effective charge on the protein network can have detrimental effects on bulk tissue strain and thus fracture toughness. However, in light of the updated Figures 8A and 8B, these previous statements have been removed.

Another thing that is challenging to the interpretation of the results is that fact OPN bridges mineral crystals, presumably more bridges with more phosphorylation. Correct? This is the first mechanism described. In the OPN^-/-^ bone, this mechanism is gone. Is the relationship between fracture toughness of OPN^-/-^ bone and phosphorylation levels (from the 2 enzyme treatments) non-linear? Sorry, Figure 8B is not too convincing with only 4 data points.

The interpretation of the result that OPN bridges mineral crystals and more bridges are formed with more phosphorylation is correct. This mechanism explains the continuous relationship between increased phosphorylation levels and increased fracture toughness with the presence of osteopontin as shown in Figure 8A. However, in the absence of OPN there seems to be an alternative mechanism where the phosphorylation and dephosphorylation of bone matrix increases and decreases toughness in a binary/step-wise fashion as seen from the results in Figure 4 and Figure 5. In particular, we postulate that the increase in global phosphorylation of other bone matrix proteins in the absence of OPN (e.g., other SIBLING matrix proteins) may potentially result in increased protein alignment and larger interfilament distance between mineralized collagen fibrils to a detrimental degree that decreases matrix interaction, energy dissipation, and consequently fracture resistance. Taken together, these results show that phosphorylation affects bone toughness through two mechanisms. First involving increased bridge formation between OPN and mineral crystals (adhesion) and between OPN molecules (cohesion) with increasing phosphorylation, and the second involving other SIBLING proteins where increased phosphorylation results in increased protein alignment and larger interfilament distance leading to decreased matrix interaction and fracture resistance. Recent work has also shown that the characteristic fibrillar structure of collagen is lost and the collagen network is disorganized when Osteopontin is absent (Depalle et al., 2020) which is likely to affect mechanical function of OPN KO mice. Please refer to the Discussion section.